# Sport psychology and performance meta-analyses: A systematic review of the literature

**Marc Lochbaum**[1,2]*, **Elisabeth Stoner**[3], **Tristen Hefner**[3], **Sydney Cooper**[4], **Andrew M. Lane**[5], **Peter C. Terry**[6]

**1** Department of Kinesiology and Sport Management, Texas Tech University, Lubbock, Texas, United States of America, **2** Education Academy, Vytautas Magnus University, Kaunas, Lithuania, **3** Department of Psychological Sciences, Texas Tech University, Lubbock, Texas, United States of America, **4** Department of Kinesiology and Sport Management, Honors College, Texas Tech University, Lubbock, Texas, United States of America, **5** Faculty of Education, Health and Well-Being, University of Wolverhampton, Walsall, West Midlands, United Kingdom, **6** Division of Research & Innovation, University of Southern Queensland, Toowoomba, Queensland, Australia

* marc.lochbaum@ttu.edu

**Data Availability Statement:** All relevant data are within the paper.

**Funding:** The author(s) received no specific funding for this work.

## Abstract

Sport psychology as an academic pursuit is nearly two centuries old. An enduring goal since inception has been to understand how psychological techniques can improve athletic performance. Although much evidence exists in the form of meta-analytic reviews related to sport psychology and performance, a systematic review of these meta-analyses is absent from the literature. We aimed to synthesize the extant literature to gain insights into the overall impact of sport psychology on athletic performance. Guided by the PRISMA statement for systematic reviews, we reviewed relevant articles identified via the EBSCOhost interface. Thirty meta-analyses published between 1983 and 2021 met the inclusion criteria, covering 16 distinct sport psychology constructs. Overall, sport psychology interventions/variables hypothesized to enhance performance (e.g., cohesion, confidence, mindfulness) were shown to have a moderate beneficial effect ($d = 0.51$), whereas variables hypothesized to be detrimental to performance (e.g., cognitive anxiety, depression, ego climate) had a small negative effect ($d = -0.21$). The quality rating of meta-analyses did not significantly moderate the magnitude of observed effects, nor did the research design (i.e., intervention vs. correlation) of the primary studies included in the meta-analyses. Our review strengthens the evidence base for sport psychology techniques and may be of great practical value to practitioners. We provide recommendations for future research in the area.

## Introduction

Sport performance matters. Verifying its global importance requires no more than opening a newspaper to the sports section, browsing the internet, looking at social media outlets, or scanning abundant sources of sport information. Sport psychology is an important avenue through which to better understand and improve sport performance. To date, a systematic review of published sport psychology and performance meta-analyses is absent from the literature. Given the undeniable importance of sport, the history of sport psychology in academics since 1830, and the global rise of sport psychology journals and organizations, a comprehensive

**Competing interests:** The authors have declared that no competing interests exist.

systematic review of the meta-analytic literature seems overdue. Thus, we aimed to consolidate the existing literature and provide recommendations for future research.

## The development of sport psychology

The history of sport psychology dates back nearly 200 years. Terry [1] cites Carl Friedrich Koch's (1830) publication titled [in translation] *Calisthenics from the Viewpoint of Dietetics and Psychology* [2] as perhaps the earliest publication in the field, and multiple commentators have noted that sport psychology experiments occurred in the world's first psychology laboratory, established by Wilhelm Wundt at the University of Leipzig in 1879 [1, 3]. Konrad Rieger's research on hypnosis and muscular endurance, published in 1884 [4] and Angelo Mosso's investigations of the effects of mental fatigue on physical performance, published in 1891 [5] were other early landmarks in the development of applied sport psychology research. Following the efforts of Koch, Wundt, Rieger, and Mosso, sport psychology works appeared with increasing regularity, including Philippe Tissié's publications in 1894 [6, 7] on psychology and physical training, and Pierre de Coubertin's first use of the term sport psychology in his *La Psychologie du Sport* paper in 1900 [8]. In short, the history of sport psychology and performance research began as early as 1830 and picked up pace in the latter part of the 19th century. Early pioneers, who helped shape sport psychology include Wundt, recognized as the "father of experimental psychology", Tissié, the founder of French physical education and Legion of Honor awardee in 1932, and de Coubertin who became the father of the modern Olympic movement and founder of the International Olympic Committee.

Sport psychology flourished in the early 20th century [see 1, 3 for extensive historic details]. For instance, independent laboratories emerged in Berlin, Germany, established by Carl Diem in 1920; in St. Petersburg and Moscow, Russia, established respectively by Avksenty Puni and Piotr Roudik in 1925; and in Champaign, Illinois USA, established by Coleman Griffith, also in 1925. The period from 1950–1980 saw rapid strides in sport psychology, with Franklin Henry establishing this field of study as independent of physical education in the landscape of American and eventually global sport science and kinesiology graduate programs [1]. In addition, of great importance in the 1960s, three international sport psychology organizations were established: namely, the International Society for Sport Psychology (1965), the North American Society for the Psychology of Sport and Physical Activity (1966), and the European Federation of Sport Psychology (1969). Since that time, the Association of Applied Sport Psychology (1986), the South American Society for Sport Psychology (1986), and the Asian-South Pacific Association of Sport Psychology (1989) have also been established.

The global growth in academic sport psychology has seen a large number of specialist publications launched, including the following journals: *International Journal of Sport Psychology* (1970), *Journal of Sport & Exercise Psychology* (1979), *The Sport Psychologist* (1987), *Journal of Applied Sport Psychology* (1989), *Psychology of Sport and Exercise* (2000), *International Journal of Sport and Exercise Psychology* (2003), *Journal of Clinical Sport Psychology* (2007), *International Review of Sport and Exercise Psychology* (2008), *Journal of Sport Psychology in Action* (2010), *Sport, Exercise, and Performance Psychology* (2014), and the *Asian Journal of Sport & Exercise Psychology* (2021).

In turn, the growth in journal outlets has seen sport psychology publications burgeon. Indicative of the scale of the contemporary literature on sport psychology, searches completed in May 2021 within the Web of Science Core Collection, identified 1,415 publications on goal setting and sport since 1985; 5,303 publications on confidence and sport since 1961; and 3,421 publications on anxiety and sport since 1980. In addition to academic journals,

several comprehensive edited textbooks have been produced detailing sport psychology developments across the world, such as Hanrahan and Andersen's (2010) *Handbook of Applied Sport Psychology* [9], Schinke, McGannon, and Smith's (2016) *International Handbook of Sport Psychology* [10], and Bertollo, Filho, and Terry's (2021) *Advancements in Mental Skills Training* [11] to name just a few. In short, sport psychology is global in both academic study and professional practice.

## Meta-analysis in sport psychology

Several meta-analysis guides, computer programs, and sport psychology domain-specific primers have been popularized in the social sciences [12, 13]. Sport psychology academics have conducted quantitative reviews on much studied constructs since the 1980s, with the first two appearing in 1983 in the form of Feltz and Landers' meta-analysis on mental practice [14], which included 98 articles dating from 1934, and Bond and Titus' cross-disciplinary meta-analysis on social facilitation [15], which summarized 241 studies including Triplett's (1898) often-cited study of social facilitation in cycling [16]. Although much meta-analytic evidence exists for various constructs in sport and exercise psychology [12] including several related to performance [17], the evidence is inconsistent. For example, two meta-analyses, both ostensibly summarizing evidence of the benefits to performance of task cohesion [18, 19], produced very different mean effects ($d$ = .24 vs $d$ = 1.00) indicating that the true benefit lies somewhere in a wide range from small to large. Thus, the lack of a reliable evidence base for the use of sport psychology techniques represents a significant gap in the knowledge base for practitioners and researchers alike. A comprehensive systematic review of all published meta-analyses in the field of sport psychology has yet to be published.

## Purpose and aim

We consider this review to be both necessary and long overdue for the following reasons: (a) the extensive history of sport psychology and performance research; (b) the prior publication of many meta-analyses summarizing various aspects of sport psychology research in a piecemeal fashion [12, 17] but not its totality; and (c) the importance of better understanding and hopefully improving sport performance via the use of interventions based on solid evidence of their efficacy. Hence, we aimed to collate and evaluate this literature in a systematic way to gain improved understanding of the impact of sport psychology variables on sport performance by construct, research design, and meta-analysis quality, to enhance practical knowledge of sport psychology techniques and identify future lines of research inquiry. By systematically reviewing all identifiable meta-analytic reviews linking sport psychology techniques with sport performance, we aimed to evaluate the strength of the evidence base underpinning sport psychology interventions.

## Materials and methods

This systematic review of meta-analyses followed the Preferred Reporting Items for Systematic Reviews and Meta-Analyses (PRISMA) guidelines [20]. We did not register our systematic review protocol in a database. However, we specified our search strategy, inclusion criteria, data extraction, and data analyses in advance of writing our manuscript. All details of our work are available from the lead author. Concerning ethics, this systematic review received a waiver from Texas Tech University Human Subject Review Board as it concerned archival data (i.e., published meta-analyses).

## Eligibility criteria

Published meta-analyses were retained for extensive examination if they met the following inclusion criteria: (a) included meta-analytic data such as mean group, between or within-group differences or correlates; (b) published prior to January 31, 2021; (c) published in a peer-reviewed journal; (d) investigated a recognized sport psychology construct; and (e) meta-analyzed data concerned with sport performance. There was no language of publication restriction. To align with our systematic review objectives, we gave much consideration to study participants and performance outcomes. Across multiple checks, all authors confirmed study eligibility. Three authors (ML, AL, and PT) completed the final inclusion assessments.

## Information sources

Authors searched electronic databases, personal meta-analysis history, and checked with personal research contacts. Electronic database searches occurred in EBSCOhost with the following individual databases selected: APA PsycINFO, ERIC, Psychology and Behavioral Sciences Collection, and SPORTDiscus. An initial search concluded October 1, 2020. ML, AL, and PT rechecked the identified studies during the February–March, 2021 period, which resulted in the identification of two additional meta-analyses [21, 22].

## Search protocol

ML and ES initially conducted independent database searches. For the first search, ML used the following search terms: sport psychology with meta-analysis or quantitative review and sport and performance or sport* performance. For the second search, ES utilized a sport psychology textbook and used the chapter title terms (e.g., goal setting). In EBSCOhost, both searches used the advanced search option that provided three separate boxes for search terms such as box 1 (sport psychology), box 2 (meta-analysis), and box 3 (performance). Specific details of our search strategy were:

Search by ML:

- sport psychology, meta-analysis, sport and performance

- sport psychology, meta-analysis or quantitative review, sport* performance

- sport psychology, quantitative review, sport and performance

- sport psychology, quantitative review, sport* performance

Search by ES:

- mental practice or mental imagery or mental rehearsal and sports performance and meta-analysis

- goal setting and sports performance and meta-analysis

- anxiety and stress and sports performance and meta-analysis

- competition and sports performance and meta-analysis

- diversity and sports performance and meta-analysis

- cohesion and sports performance and meta-analysis

- imagery and sports performance and meta-analysis

- self-confidence and sports performance and meta-analysis

- concentration and sports performance and meta-analysis

- athletic injuries and sports performance and meta-analysis

- overtraining and sports performance and meta-analysis

- children and sports performance and meta-analysis

The following specific search of the EBSCOhost with SPORTDiscus, APA PsycINFO, Psychology and Behavioral Sciences Collection, and ERIC databases, returned six results from 2002–2020, of which three were included [18, 19, 23] and three were excluded because they were not meta-analyses.

- Box 1 cohesion

- Box 2 sports performance

- Box 3 meta-analysis

## Study selection

As detailed in the PRISMA flow chart (Fig 1) and the specified inclusion criteria, a thorough study selection process was used. As mentioned in the search protocol, two authors (ML and ES) engaged independently with two separate searches and then worked together to verify the selected studies. Next, AL and PT examined the selected study list for accuracy. ML, AL, and PT, whilst rating the quality of included meta-analyses, also re-examined all selected studies to verify that each met the predetermined study inclusion criteria. Throughout the study selection process, disagreements were resolved through discussion until consensus was reached.

## Data extraction process

Initially, ML, TH, and ES extracted data items 1, 2, 3 and 8 (see Data items). Subsequently, ML, AL, and PT extracted the remaining data (items 4–7, 9, 10). Checks occurred during the extraction process for potential discrepancies (e.g., checking the number of primary studies in a meta-analysis). It was unnecessary to contact any meta-analysis authors for missing information or clarification during the data extraction process because all studies reported the required information. Across the search for meta-analyses, all identified studies were reported in English. Thus, no translation software or searching out a native speaker occurred. All data extraction forms (e.g., data items and individual meta-analysis quality) are available from the first author.

## Data items

To help address our main aim, we extracted the following information from each meta-analysis: (1) author(s); (2) publication year; (3) construct(s); (4) intervention based meta-analysis (yes, no, mix); (5) performance outcome(s) description; (6) number of studies for the performance outcomes; (7) participant description; (8) main findings; (9) bias correction method/ results; and (10) author(s) stated conclusions. For all information sought, we coded missing information as not reported.

## Individual meta-analysis quality

ML, AL, and PT independently rated the quality of individual meta-analysis on the following 25 points found in the PRISMA checklist [20]: title; abstract structured summary; introduction rationale, objectives, and protocol and registration; methods eligibility criteria, information

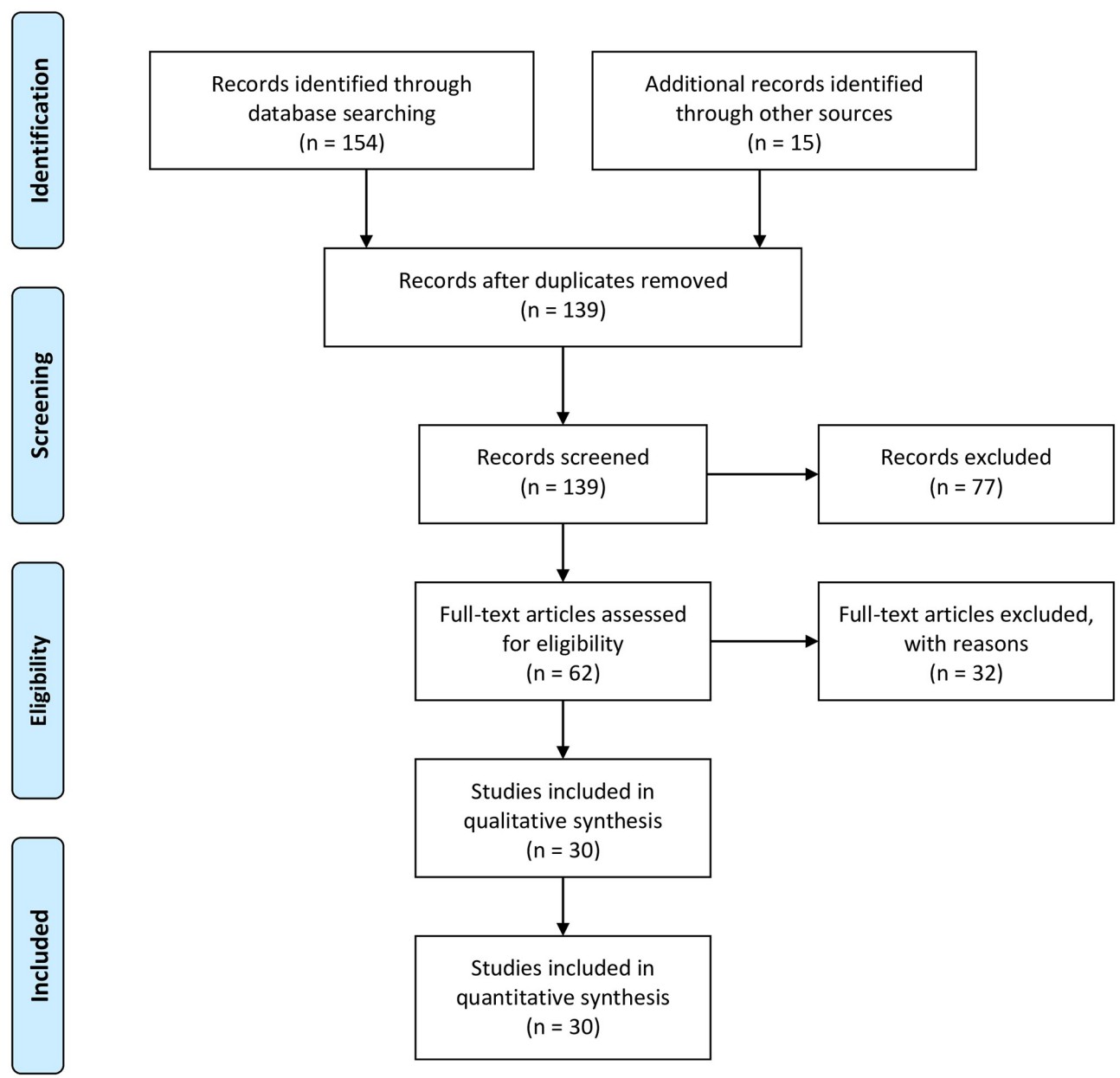

**Fig 1. PRISMA flow chart for the identification of the included studies.**

sources, search, study selection, data collection process, data items, risk of bias of individual studies, summary measures, synthesis of results, and risk of bias across studies; results study selection, study characteristics, risk of bias within studies, results of individual studies, synthesis of results, and risk of bias across studies; discussion summary of evidence, limitations, and conclusions; and funding. All meta-analyses were rated for quality by two coders to facilitate inter-coder reliability checks, and the mean quality ratings were used in subsequent analyses. One author (PT), having completed his own ratings, received the incoming ratings from ML and AL and ran the inter-coder analysis. Two rounds of ratings occurred due to discrepancies for seven meta-analyses, mainly between ML and AL. As no objective quality categorizations (i.e., a point system for grouping meta-analyses as poor, medium, good) currently exist, each

meta-analysis was allocated a quality score of up to a maximum of 25 points. All coding records are available upon request.

## Planned methods of analysis

Several preplanned methods of analysis occurred. We first assessed the mean quality rating of each meta-analysis based on our 25-point PRISMA-based rating system. Next, we used a median split of quality ratings to determine whether standardized mean effects (SMDs) differed by the two formed categories, higher and lower quality meta-analyses. Meta-analysis authors reported either of two different effect size metrics (i.e., *r* and SMD); hence we converted all correlational effects to SMD (i.e., Cohen's *d*) values using an online effect size calculator (www.polyu.edu.hk/mm/effectsizefaqs/calculator/calculator.html). We interpreted the meaningfulness of effects based on Cohen's interpretation [24] with 0.20 as small, 0.50 as medium, 0.80 as large, and 1.30 as very large. As some psychological variables associate negatively with performance (e.g., confusion [25], cognitive anxiety [26]) whereas others associate positively (e.g., cohesion [23], mental practice [14]), we grouped meta-analyses according to whether the hypothesized effect with performance was positive or negative, and summarized the overall effects separately. By doing so, we avoided a scenario whereby the demonstrated positive and negative effects canceled one another out when combined. The effect of somatic anxiety on performance, which is hypothesized to follow an inverted-U relationship, was categorized as neutral [35]. Last, we grouped the included meta-analyses according to whether the primary studies were correlational in nature or involved an intervention and summarized these two groups of meta-analyses separately.

## Results

### Study characteristics

Table 1 contains extracted data from 30 meta-analyses meeting the inclusion criteria, dating from 1983 [14] to 2021 [21]. The number of primary studies within the meta-analyses ranged from three [27] to 109 [28]. In terms of the description of participants included in the meta-analyses, 13 included participants described simply as athletes, whereas other meta-analyses identified a mix of elite athletes (e.g., professional, Olympic), recreational athletes, college-aged volunteers (many from sport science departments), younger children to adolescents, and adult exercisers. Of the 30 included meta-analyses, the majority (*n* = 18) were published since 2010. The decadal breakdown of meta-analyses was 1980–1989 (*n* = 1 [14]), 1990–1999 (*n* = 6 [29–34]), 2000–2009 (*n* = 5 [23, 25, 26, 35, 36]), 2010–2019 (*n* = 12 [18, 19, 22, 27, 37–43, 48]), and 2020–2021 (*n* = 6 [21, 28, 44–47]).

As for the constructs covered, we categorized the 30 meta-analyses into the following areas: mental practice/imagery [14, 29, 30, 42, 46, 47], anxiety [26, 31, 32, 35], confidence [26, 35, 36], cohesion [18, 19, 23], goal orientation [22, 44, 48], mood [21, 25, 34], emotional intelligence [40], goal setting [33], interventions [37], mindfulness [27], music [28], neurofeedback training [43], perfectionism [39], pressure training [45], quiet eye training [41], and self-talk [38]. Multiple effects were generated from meta-analyses that included more than one construct (e.g., tension, depression, etc. [21]; anxiety and confidence [26]). In relation to whether the meta-analyses included in our review assessed the effects of a sport psychology intervention on performance or relationships between psychological constructs and performance, 13 were intervention-based, 14 were correlational, two included a mix of study types, and one included a large majority of cross-sectional studies (Table 1).

A wide variety of performance outcomes across many sports was evident, such as golf putting, dart throwing, maximal strength, and juggling; or categorical outcomes such as win/loss

**Table 1. Citation, construct, outcome(s), number of primary studies, type of participants, main findings, bias correction, and author conclusions for 30 meta-analyses.**

| Study | Construct | Outcome(s) | Studies | Participants | Main Findings | Bias Checked | Author Conclusions |
|---|---|---|---|---|---|---|---|
| Beedie et al. [25] | Mood | Sport performance (e.g., Olympic selection, win/ loss). Athletic achievement (e.g., elite vs. club athletes) | 16, 13 | Athletes | Performance SMD = .31 ± 12; Achievement SMD = .10 ± .07 | NR | "Mood profiles have utility in predicting athletic performance but not level of achievement." |
| Brown & Fletcher [37] | Psychosocial Interventions | Variety of sport outcomes—such as bowling, tennis, high jump and variety of technical tasks such as Australian football set shot, golf shot, and volleyball pass | 35, 8 | Athletes | Post-test SMD = 0.57, [0.22–0.92]; Follow-up SMD = 1.16 [0.25–2.08] | Yes | "Psychological and psychosocial interventions have a moderate positive effect on sport performance, and this effect may last at least a month following the end of the intervention." |
| Bühlmayer et al. [27] | Mindfulness | Shooting and dart throwing | 3 | Athletes | SMD = 1.35 [.61–2.09] | NR | "Furthermore, physiological and psychological surrogates improved to a meaningful extent following mindfulness practice, as well as performance outcomes in shooting and dart throwing." |
| Carron et al. [23] | Cohesion | Performance outcomes from interactive (e.g., ice hockey) and coactive (i.e., rowing) teams | 46 | Athletes | All SMD = .65 ± .95; Correlational SMD = .69 ± .91; Experimental SMD = .40 ± .46 | Yes | "The overall effect size using all operational measures of cohesion showed that a significant moderate to large cohesion-performance relationship is present for sport teams." |
| Castaño et al. [18] | Cohesion | Vaguely described. Win/ loss ratio a stated example. | 13, 6 | Athletes | Social Cohesion r = .16 ± .22; Task Cohesion r = .12 ± .12 | Yes | "Results showed that the task cohesion-performance relationship is different in a sports setting from a business setting, with the latter showing a stronger effect." |
| Craft et al. [35] | Anxiety, Self-Confidence | Variety of athletic performance outcomes | 29 | Predominantly athletes with college PE students | Cognitive Anxiety r = .01 [-.03, .04]; Somatic Anxiety r = -.03 [-.08, .01]; Self-Confidence r = .25 [.20, .28] | NR | "Relationships among cognitive anxiety, somatic anxiety, self-confidence and performance appeared weak." |
| Driskell et al. [29] | Mental Practice | Physical (muscular strength, endurance, coordination) and cognitive (perceptual input, mental operations, output and response) | 35 | Not described, most likely college-aged mix of novice and experienced with performance task | r = .255 ± .527 | NR | "...mental practice has a positive and significant effect on performance, and the effectiveness of mental practice was moderated by the type of task, the retention interval between practice and performance, and the length or duration of the mental practice intervention." |
| Feltz & Landers [14] | Mental Practice | Motor (juggling, basketball free throws), strength (cable tensiometer, sit-ups), and cognitive (card sorting, dial-a-maze) tasks | 60 | Primarily college-aged, mostly novice to performance task | SMD = .48 ± .67 | Yes | "...mentally practicing a motor skill influences performance somewhat better than no practice at all....studies employing cognitive tasks had larger average effect sizes than motor or strength tasks and that published studies had larger average effect sizes than unpublished studies." |

*(Continued)*

**Table 1.** (*Continued*)

| Study | Construct | Outcome(s) | Studies | Participants | Main Findings | Bias Checked | Author Conclusions |
|---|---|---|---|---|---|---|---|
| Filho et al. [19] | Cohesion | Majority objective athletic outcomes | 17 | Recreational, interscholastic, collegiate, and professional athletes | Overall Cohesion r = 0.34 [.26, .34]; Task Cohesion r = 0.45 [.39, .46]; Social Cohesion r = 0.11 [.03, .22]. | Yes | "Results revealed a statistically significant moderate relationship between overall cohesion and performance, large relationship between task cohesion and performance, and small relationship between social cohesion and performance." |
| Harwood et al. [22] | Achievement goal climate | Win/loss percentage, cardiovascular fitness, evaluated skill level, one-mile time | 11 | Primarily students, student-athletes in activity settings | Task Climate r = .24 CI [.14, .35]; Ego Climate r = -.08 CI [-.15, -.02] | Yes | "Perceptions of a task or mastery climate were consistently associated with a range of adaptive motivational outcomes including. . .objective performance. . . Perceptions of an ego or performance climate. . . (objective performance outcomes not mentioned)." |
| Hatzigeorgiadis et al. [38] | Self-talk | Classified as fine and gross motor skills | 32 | Mostly students and beginning athletes with some experienced athletes | SMD = .48 [.38, .58] | NR | "The results of this study establish the effectiveness of self-talk in sport, encourage the use of self-talk as a strategy to facilitate learning and enhance performance, and provide new research directions." |
| Hill et al. [39] | Perfectionism | Individual sport outcomes (e.g., triathlon race times, mile time) | 6 | Predominantly competing athletes with some sport science students | Perfectionistic Strivings r = .23 [.11, .35]; Perfectionistic Concerns r = .06 [-.01, .14] | Yes | "Random effects models revealed that perfectionistic strivings displayed. . .a small-to-medium relationship with better performance. By contrast, perfectionistic concerns. . .were unrelated to performance |
| Hinshaw [30] | Mental Practice | Predominantly athletic (e.g., dart throwing), strength (e.g., bench press), or laboratory (e.g., stabilometer) skills | 21 | Mix elementary to college aged, novice and experienced | Overall SMD = .68 ± .11, Internal Perspective SMD = 1.34 ± .32, External Perspective SMD = .80 ± .12 | NR | "There is a significant benefit to performance of using mental practice over no practice . . . use of "internal" imagery produced a larger average effect size than use of "external" imagery |
| Ivarsson et al. [44] | Achievement goals | Progression (e.g., higher level team), football statistics (e.g., goals, assists) | 11 | Athletes | Ego Goal SMD = .06 [-03, .14]; Task Goal SMD = .28 [.07, .50], Task-Oriented Coping SMD = .20 [.11, .28] | NR | "Psychological factors investigated showed small effects on future football performance, however, there was overall uncertainty in this evidence due to various sources of bias in the included studies." |
| Jokela & Hanin [31] | Anxiety | Sport performance (criterion referenced, self-referenced, or subjectively rated) | 19 | Athletes | Overall SMD = .44 [.32, .55], Criterion-Referenced SMD = .61 [.49, .73], Self-Referenced SMD = .27 [-.09, .63] | NR | "The performance of athletes who were within their individually optimal zones were almost one half a standard deviation unit better than of athletes who were outside their zones" |
| Kleine [32] | Anxiety | Performance in 20 + sports operationalized in multiple ways | 50 | Predominantly athletes with general students in PE | Overall r = -.19, Women r = -.23 [-.72, .27], Men r = -.12 [-.38, .14] | NR | "Anxiety and sport performance correlated consistently negatively" |

(*Continued*)

**Table 1.** (Continued)

| Study | Construct | Outcome(s) | Studies | Participants | Main Findings | Bias Checked | Author Conclusions |
|-------|-----------|------------|---------|--------------|---------------|--------------|--------------------|
| Kopp & Jekauc [40] | Emotional Intelligence | Competition statistics, level of achievement, subjective assessment, and physical parameters (e.g., maximal voluntary contraction) | 21 | Athletes | r = .16 [.11, .22] | Yes | "The meta-analysis...found a small but significant relationship between EI and sports performance...Overall, the result is encouraging regarding the value of EI as a possible predictor in sports performance." |
| Kyllo & Landers [33] | Goal Setting | Sport, exercise, or motor performance | 36 | Mix of pre-teens, adolescents, and college-aged students | SMD = .34 ± .03 | NR | "Goal setting may be improved by specifying goals in absolute (i.e., outcome), terns (ES = 0.93), by setting short-term and long-term goals (ES = 0.48), by allowing individuals to participate in setting goals (ES = 0.62), and by making the goals public (ES = 0.79). The effectiveness of goal setting in improving sport and exercise performance appears to be well established." |
| Lebeau et al. [41] | Quiet Eye | Self-paced sport performance (e.g., golf-putting, basketball free throws) | 9 | Not specifically stated, most likely mix athletes in their sports and college aged volunteers | SMD = .84 [.61, 1.06]. Bias-corrected: SMD = .69 [.58, .80] | Yes | "The results signify the QE period as a key perceptual-cognitive variable affecting performance. QE is a marginally significant predictor of performance across intervention studies" |
| Lochbaum et al. [21] | Mood | Achievement (e.g., made Olympic team) and game statistics in variety of sports | 25 | Athletes | SMD Tension = −.21 [-.51, .09], Depression −.43 [-.75, -.11], Anger −.08 [-.15, .30], Vigor .38 [.15, .60], Fatigue −.13 [-.46, .20], Confusion −.41 [-.76, -.06], Total Mood Disturbance −.53 [-1.14, .07] | Yes | Measured before performance, most of the POMS scales and TMD are reliable predictors of sport performance in competitive athletes across a wide variety of sports and athletic performance outcomes. Morgan's (1980, 1985) mental health model or iceberg profile minus anger is still a viable method for understanding and improving athletic performances. |
| Lochbaum & Gottardy [48] | Achievement Goals | Achievements, closed-skill task (e.g., putting), and physical performance (e.g., fitness test) | 17 | Athletes and non-athletes (e.g., college student volunteers) | Performance Approach Goal SMD = .38 [.22, .54], Performance Avoidance Goal SMD = -.15 [-.30, 0], Mastery Approach Goal SMD = .38 [.30, .46], Mastery Avoidance Goal SMD = -.11 [-.22, .01], Performance Goal Contrast SMD = .74 [.52, .97] | NR | "The performance goal contrast holds value for sport performance research. Contrary to approach-avoidance predictions, the mastery-approach goal and performance effect size was significant and of equal magnitude as the performance approach goal and performance effect size." |
| Low et al. [45] | Pressure Training | Mostly self-paced skills in several sport contexts (e.g., golf-putting, basketball free throws, dart throwing) | 10 | Novices and athletes | SMD = .72 [.45, 1.00] | Yes | "Results suggest coaches should create pressurized training environments rather than relying on greater amounts of training to help performers adjust to pressure." |

(*Continued*)

**Table 1.** (Continued)

| Study | Construct | Outcome(s) | Studies | Participants | Main Findings | Bias Checked | Author Conclusions |
|---|---|---|---|---|---|---|---|
| Moritz et al. [36] | Self-Efficacy | Subjective, objective or self-rated sport performance | 45 | Not specifically stated, most likely mix athletes in their sports and college aged volunteers | r = .38 [.35, .41] | NR | "The largest correlations were obtained for those studies that subjectively assessed performance (r = .47), followed by self-report (r = .44) and objective performance (r = .34)." |
| Paravlic et al. [42] | Mental Practice | Maximal voluntary strength | 13 | Healthy adults | SMD = .72 [.42, 1.02] | Yes | ". . .compared to a no-exercise control group of healthy adults, motor imagery practice increases maximal voluntary strength." |
| Rowley et al. [34] | Mood | Sport performance coded as, for example, personal best, ranking, selection for team, winning/losing, or subjective assessment. | 33 | Athletes | SMD = .15 ± .89 | Yes | ". . .successful athletes possess a mood profile slightly more positive than less successful athletes." |
| Simonsmeier et al. [46] | Imagery | Performance in 10 sports, including archery, figure skating, gymnastics, and soccer. | 55 | Athletes | SMD = .47 [.30, .63] | Yes | "Imagery interventions significantly enhanced motor performance." |
| Terry et al. [28] | Music | Objective performance (e.g., time, distance, speed, power, repetitions) in a wide variety of sports and physical activities | 109 | Athletes and exercisers | SMD Performance = .31 [.25, .36] | Yes | "Music can enhance performance" |
| Toth [47] | Mental Practice | Performance quantified according to distance (e.g., distance from the target), time (e.g., time to complete a task), or other (e.g., idiosyncratic scoring system). | 37 | Not specifically stated, most likely mix athletes in their sports and college aged volunteers | r = .24 [.12, .28] | Yes | "Mental practice has a small but significant positive effect on performance." |
| Woodman & Hardy [26] | Anxiety, Self-Confidence | Performance in 20 + sports | 48 | Athletes | Cognitive Anxiety r = -.10, Self-Confidence r = .24 | Yes | ". . .both cognitive anxiety and self-confidence are significantly related to competitive sport performance." |
| Xiang et al. [43] | Neurofeedback Training (NFT) | Performance in self-paced sports, including archery, golf, gymnastics, shooting, and swimming. | 10 | Athletes | SMD = .65 [.28, 1.03] | NR | "NFT significantly improved the sport performance." |

and Olympic team selection. Given the extensive list of performance outcomes and the incomplete descriptions provided in some meta-analyses, a clear categorization or count of performance types was not possible. Sufficient to conclude, researchers utilized many performance outcomes across a wide range of team and individual sports, motor skills, and strength and aerobic tasks.

### Effect size data and bias correction

To best summarize the effects, we transformed all correlations to SMD values (i.e., Cohen's *d*). Across all included meta-analyses shown in Table 2 and depicted in Fig 2, we identified 61 effects. Having corrected for bias, effect size values were assessed for meaningfulness [24],

**Table 2. Citation, quality metrics, study design, hypothesized performance effect, construct, standardized mean difference (SMD) and bias corrected values for 30 meta-analyses.**

| Citation | Quality Score | Quality Category | Interventions | Hypothesized Direction | Construct | SMD | SMD Bias Corrected |
|---|---|---|---|---|---|---|---|
| Beedie et al. [25] | 15.5 | Lower | No | Negative | Tension | -0.25 | -0.25 |
| | | | | Negative | Depression | -0.34 | -0.34 |
| | | | | Negative | Anger | -0.27 | -0.27 |
| | | | | Positive | Vigor | 0.47 | 0.47 |
| | | | | Negative | Fatigue | -0.13 | -0.13 |
| | | | | Negative | Confusion | -0.40 | -0.40 |
| | | | | Negative | Tension | -0.14 | -0.14 |
| | | | | Negative | Depression | 0.06 | 0.06 |
| | | | | Negative | Anger | -0.02 | -0.02 |
| | | | | Positive | Vigor | 0.22 | 0.22 |
| | | | | Negative | Fatigue | -0.04 | -0.04 |
| | | | | Negative | Confusion | -0.11 | -0.11 |
| Brown & Fletcher [37] | 23.5 | Higher | Yes | Positive | Psychosocial Interventions | 0.57 | 0.57 |
| Bühlmayer et al. [27] | 24.0 | Higher | No | Positive | Mindfulness | 1.35 | 1.35 |
| Carron et al. [23] | 13.5 | Lower | Mix | Positive | Cohesion | 0.65 | 0.65 |
| Castaño et al. [18] | 19.0 | Lower | No | Positive | Social cohesion | 0.32 | 0.32 |
| | | | | Positive | Task cohesion | 0.24 | 0.24 |
| Craft et al. [35] | 13.0 | Lower | No | Negative | Cognitive anxiety | 0.02 | 0.02 |
| | | | | Neutral | Somatic anxiety | -0.06 | -0.06 |
| | | | | Positive | Confidence | 0.51 | 0.51 |
| Driskell et al. [29] | 13.5 | Lower | Yes | Positive | Mental practice | 0.51 | 0.51 |
| Feltz & Landers [14] | 13.0 | Lower | Yes | Positive | Mental practice | 0.48 | 0.48 |
| Filho et al. [19] | 20.5 | Lower | Vast majority not | Positive | Overall cohesion | 0.72 | 0.72 |
| | | | | Positive | Task cohesion | 1.00 | 1.00 |
| | | | | Positive | Social cohesion | 0.22 | 0.22 |
| Harwood et al. [22] | 23.0 | Higher | No | Positive | Task climate | 0.49 | 0.49 |
| | | | | Negative | Ego climate | -0.16 | -0.16 |
| Hatzigeorgiadis et al. [38] | 15.0 | Lower | Yes | Positive | Self-talk | 0.48 | 0.48 |
| Hill et al. [39] | 21.5 | Higher | No | Positive | Perfectionistic strivings | 0.47 | 0.53 |
| | | | | Negative | Perfectionistic concerns | 0.12 | 0.12 |
| Hinshaw [30] | 13.5 | Lower | Yes | Positive | Mental practice | 0.68 | 0.68 |
| Ivarsson et al. [44] | 23.5 | Higher | No | Negative | Ego goal | 0.06 | 0.06 |
| | | | | Positive | Task goal | 0.28 | 0.28 |
| | | | | Positive | Task-oriented coping | 0.20 | 0.20 |
| Jokela & Hanin [31] | 18.0 | Lower | No | Positive | Anxiety | 0.44 | 0.44 |
| Kleine [32] | 14.5 | Lower | No | Negative | Anxiety | -0.38 | -0.38 |
| Kopp & Jekauc [40] | 23.5 | Higher | No | Positive | Emotional intelligence | 0.32 | 0.32 |
| Kyllo & Landers [33] | 16.0 | Lower | Yes | Positive | Goal Setting | 0.34 | 0.34 |
| Lebeau et al. [41] | 22.0 | Higher | Yes | Positive | Quiet Eye | 0.84 | 0.69 |
| Lochbaum et al. [21] | 25.0 | Higher | No | Negative | Tension | -0.21 | -0.47 |
| | | | | Negative | Depression | -0.43 | -0.64 |
| | | | | Negative | Anger | -0.08 | -0.08 |
| | | | | Positive | Vigor | 0.38 | 0.44 |
| | | | | Negative | Fatigue | -0.13 | -0.34 |
| | | | | Negative | Confusion | -0.41 | -0.41 |
| | | | | Negative | Total mood disturbance | -0.53 | -0.84 |

*(Continued)*

**Table 2.** (Continued)

| Citation | Quality Score | Quality Category | Interventions | Hypothesized Direction | Construct | SMD | SMD Bias Corrected |
|---|---|---|---|---|---|---|---|
| Lochbaum & Gottardy [48] | 21.5 | Higher | Mix | Positive | Performance approach goal | 0.38 | 0.38 |
| | | | | Negative | Performance avoidance goal | -0.15 | -0.15 |
| | | | | Positive | Mastery approach goal | 0.38 | 0.38 |
| | | | | Negative | Mastery avoidance goal | 0.11 | 0.11 |
| | | | | Positive | Performance goal contrast | 0.74 | 0.74 |
| Low et al. [45] | 22.5 | Higher | Yes | Positive | Pressure training | 0.72 | 0.72 |
| Moritz et al. [36] | 18.0 | Lower | No | Positive | Self-efficacy | 0.82 | 0.82 |
| Paravlic et al. [42] | 23.5 | Higher | Yes | Positive | Mental practice | 0.72 | 0.72 |
| Rowley et al. [34] | 13.0 | Lower | No | Positive | Mood | 0.15 | 0.15 |
| Simonsmeier et al. [46] | 21.5 | Higher | Yes | Positive | Mental practice | 0.47 | 0.47 |
| Terry et al. [28] | 23.5 | Higher | Yes | Positive | Music | 0.31 | 0.31 |
| Toth et al. [47] | 21.5 | Higher | Yes | Positive | Mental practice | 0.49 | 0.26 |
| Woodman & Hardy [26] | 16.0 | Lower | No | Negative | Cognitive anxiety | -0.20 | -0.20 |
| | | | | Positive | Confidence | 0.49 | 0.49 |
| Xiang et al. [43] | 22.5 | Higher | Yes | Positive | Neurofeedback training | 0.65 | 0.65 |

which resulted in 15 categorized as negligible ($< \pm 0.20$), 29 as small ($\pm 0.20$ to $< 0.50$), 13 as moderate ($\pm 0.50$ to $< 0.80$), 2 as large ($\pm 0.80$ to $< 1.30$), and 1 as very large ($\geq 1.30$).

## Study quality rating results and summary analyses

Following our PRISMA quality ratings, intercoder reliability coefficients were initially .83 (ML, AL), .95 (ML, PT), and .90 (AL, PT), with a mean intercoder reliability coefficient of .89. To achieve improved reliability (i.e., $r_{mean} > .90$), ML and AL re-examined their ratings. As a result, intercoder reliability increased to .98 (ML, AL), .96 (ML, PT), and .92 (AL, PT); a mean intercoder reliability coefficient of .95. Final quality ratings (i.e., the mean of two coders) ranged from 13 to 25 ($M = 19.03 \pm 4.15$). Our median split into higher ($M = 22.83 \pm 1.08$, range 21.5–25, $n = 15$) and lower ($M = 15.47 \pm 2.42$, range 13–20.5, $n = 15$) quality groups produced significant between-group differences in quality ($F_{1,28} = 115.62$, $p < .001$); hence, the median split met our intended purpose. The higher quality group of meta-analyses were published from 2015–2021 (median 2018) and the lower quality group from 1983–2014 (median 2000). It appears that meta-analysis standards have risen over the years since the PRISMA criteria were first introduced in 2009. All data for our analyses are shown in Table 2.

Table 3 contains summary statistics with bias-corrected values used in the analyses. The overall mean effect for sport psychology constructs hypothesized to have a positive impact on performance was of moderate magnitude ($d = 0.51$, 95% CI = 0.42, 0.58, $n = 36$). The overall mean effect for sport psychology constructs hypothesized to have a negative impact on performance was small in magnitude ($d = -0.21$, 95% CI -0.31, -0.11, $n = 24$). In both instances, effects were larger, although not significantly so, among meta-analyses of higher quality compared to those of lower quality. Similarly, mean effects were larger but not significantly so, where reported effects in the original studies were based on interventional rather than correlational designs. This trend only applied to hypothesized positive effects because none of the original studies in the meta-analyses related to hypothesized negative effects used interventional designs.

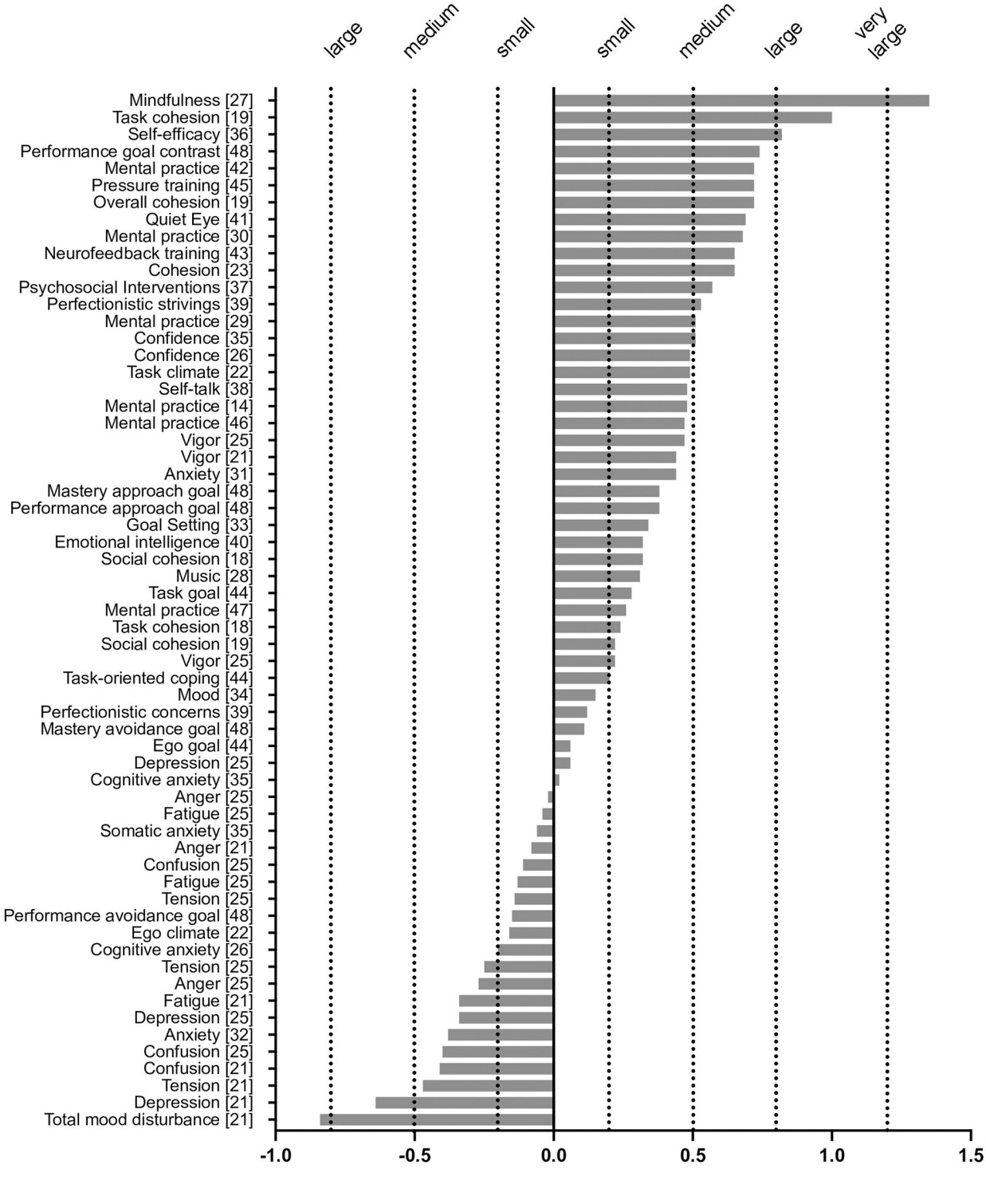

**Fig 2. Standardized mean difference (SMD) values by meta-analysis construct with meaningfulness categories.**

**Table 3. Summary statistics for meta-analyses grouped by quality, hypothesized direction of effects, and inclusion of intervention/correlational studies.**

| Source | Group | Quality | k | Mean SMD ± SE | t | p |
|---|---|---|---|---|---|---|
| Hypothesized Effect | Positive | Overall | 36 | .51 ± .04 | 11.18 | < .001 |
| | | Higher | 21 | .54 ± .06 | 1.11 | N.S. |
| | | Lower | 15 | .45 ± .05 | | |
| | Negative | Overall | 24 | -.21 ± .05 | | |
| | | Higher | 11 | -.25 ± .09 | 0.87 | N.S. |
| | | Lower | 13 | -.17 ± .04 | | |
| Inclusion of intervention | Positive | Overall | 29 | .49 ± .04 | | |
| | | Intervention | 13 | .53 ± .04 | 0.83 | N.S. |
| | | Correlation | 16 | .45 ± .07 | | |
| | Negative | Overall | 22 | -.22 ± .05 | | |
| | | Intervention | 0 | | n/a | |
| | | Correlation | 22 | -.22 ± .05 | | |

Note. $k$ = number of effects, N.S. = non-significant, n/a = not applicable.

## Discussion

In this systematic review of meta-analyses, we synthesized the available evidence regarding effects of sport psychology interventions/constructs on sport performance. We aimed to consolidate the literature, evaluate the potential for meta-analysis quality to influence the results, and suggest recommendations for future research at both the single study and quantitative review stages. During the systematic review process, several meta-analysis characteristics came to light, such as the number of meta-analyses of sport psychology interventions (experimental designs) compared to those summarizing the effects of psychological constructs (correlation designs) on performance, the number of meta-analyses with exclusively athletes as participants, and constructs featuring in multiple meta-analyses, some of which (e.g., cohesion) produced very different effect size values. Thus, although our overall aim was to evaluate the strength of the evidence base for use of psychological interventions in sport, we also discuss the impact of these meta-analysis characteristics on the reliability of the evidence.

When seen collectively, results of our review are supportive of using sport psychology techniques to help improve performance and confirm that variations in psychological constructs relate to variations in performance. For constructs hypothesized to have a positive effect on performance, the mean effect strength was moderate ($d$ = 0.51) although there was substantial variation between constructs. For example, the beneficial effects on performance of task cohesion ($d$ = 1.00) and self-efficacy ($d$ = 0.82) are large, and the available evidence base for use of mindfulness interventions suggests a very large beneficial effect on performance ($d$ = 1.35). Conversely, some hypothetically beneficial effects (2 of 36; 5.6%) were in the negligible-to-small range (0.15–0.20) and most beneficial effects (19 of 36; 52.8%) were in the small-to-moderate range (0.22–0.49). It should be noted that in the world of sport, especially at the elite level, even a small beneficial effect on performance derived from a psychological intervention may prove the difference between success and failure and hence small effects may be of great practical value. To put the scale of the benefits into perspective, an authoritative and extensively cited review of healthy eating and physical activity interventions [49] produced an overall pooled effect size of 0.31 (compared to 0.51 for our study), suggesting sport psychology interventions designed to improve performance are generally more effective than interventions designed to promote healthy living.

Among hypothetically negative effects (e.g., ego climate, cognitive anxiety, depression), the mean detrimental effect was small ($d$ = -0.21) although again substantial variation among constructs was evident. Some hypothetically negative constructs (5 of 24; 20.8%) were found to actually provide benefits to performance, albeit in the negligible range (0.02–0.12) and only two constructs (8.3%), both from Lochbaum and colleagues' POMS meta-analysis [21], were shown to negatively affect performance above a moderate level (depression: $d$ = -0.64; total mood disturbance, which incorporates the depression subscale: $d$ = -0.84). Readers should note that the POMS and its derivatives assess six specific mood dimensions rather than the mood construct more broadly, and therefore results should not be extrapolated to other dimensions of mood [50].

Mean effects were larger among higher quality than lower quality meta-analyses for both hypothetically positive ($d$ = 0.54 vs $d$ = 0.45) and negative effects ($d$ = -0.25 vs $d$ = 0.17), but in neither case were the differences significant. It is reasonable to assume that the true effects were derived from the higher quality meta-analyses, although our conclusions remain the same regardless of study quality. Overall, our findings provide a more rigorous evidence base for the use of sport psychology techniques by practitioners than was previously available, representing a significant contribution to knowledge. Moreover, our systematic scrutiny of 30 meta-analyses published between 1983 and 2021 has facilitated a series of recommendations to improve the quality of future investigations in the sport psychology area.

## Recommendations

The development of sport psychology as an academic discipline and area of professional practice relies on using evidence and theory to guide practice. Hence, a strong evidence base for the applied work of sport psychologists is of paramount importance. Although the beneficial effects of some sport psychology techniques are small, it is important to note the larger performance benefits for other techniques, which may be extremely meaningful for applied practice. Overall, however, especially given the heterogeneity of the observed effects, it would be wise for applied practitioners to avoid overpromising the benefits of sport psychology services to clients and perhaps underdelivering as a result [1].

The results of our systematic review can be used to generate recommendations for how the profession might conduct improved research to better inform applied practice. Much of the early research in sport psychology was exploratory and potential moderating variables were not always sufficiently controlled. Terry [51] outlined this in relation to the study of mood-performance relationships, identifying that physical and skills factors will very likely exert a greater influence on performance than psychological factors. Further, type of sport (e.g., individual vs. team), duration of activity (e.g., short vs. long duration), level of competition (e.g., elite vs. recreational), and performance measure (e.g., norm-referenced vs. self-referenced) have all been implicated as potential moderators of the relationship between psychological variables and sport performance [51]. To detect the relatively subtle effects of psychological effects on performance, research designs need to be sufficiently sensitive to such potential confounds. Several specific methodological issues are worth discussing.

The first issue relates to measurement. Investigating the strength of a relationship requires the measured variables to be valid, accurate and reliable. Psychological variables in the meta-analyses we reviewed relied primarily on self-report outcome measures. The accuracy of self-report data requires detailed inner knowledge of thoughts, emotions, and behavior. Research shows that the accuracy of self-report information is subject to substantial individual differences [52, 53]. Therefore, self-report data, at best, are an estimate of the measure. Measurement issues are especially relevant to the assessment of performance, and considerable

measurement variation was evident between meta-analyses. Some performance measures were more sensitive, especially those assessing physical performance relative to what is normal for the individual performer (i.e., self-referenced performance). Hence, having multiple baseline indicators of performance increases the probability of identifying genuine performance enhancement derived from a psychological intervention [54].

A second issue relates to clarifying the rationale for how and why specific psychological variables might influence performance. A comprehensive review of prerequisites and precursors of athletic talent [55] concluded that the superiority of Olympic champions over other elite athletes is determined in part by a range of psychological variables, including high intrinsic motivation, determination, dedication, persistence, and creativity, thereby identifying performance-related variables that might benefit from a psychological intervention. Identifying variables that influence the effectiveness of interventions is a challenging but essential issue for researchers seeking to control and assess factors that might influence results [49]. A key part of this process is to use theory to propose the mechanism(s) by which an intervention might affect performance and to hypothesize how large the effect might be.

A third issue relates to the characteristics of the research participants involved. Out of convenience, it is not uncommon for researchers to use undergraduate student participants for research projects, which may bias results and restrict the generalization of findings to the population of primary interest, often elite athletes. The level of training and physical conditioning of participants will clearly influence their performance. Highly trained athletes will typically make smaller gains in performance over time than novice athletes, due to a ceiling effect (i.e., they have less room for improvement). For example, consider runner A, who takes 20 minutes to run 5km one week but 19 minutes the next week, and Runner B who takes 30 minutes one week and 25 minutes the next. If we compare the two, Runner A runs faster than Runner B on both occasions, but Runner B improved more, so whose performance was better? If we also consider Runner C, a highly trained athlete with a personal best of 14 minutes, to run 1 minute quicker the following week would almost require a world record time, which is clearly unlikely. For this runner, an improvement of a few seconds would represent an excellent performance. Evidence shows that trained, highly motivated athletes may reach performance plateaus and as such are good candidates for psychological skills training. They are less likely to make performance gains due to increased training volume and therefore the impact of psychological skills interventions may emerge more clearly. Therefore, both test-retest and cross-sectional research designs should account for individual difference variables. Further, the range of individual difference factors will be context specific; for example, individual differences in strength will be more important in a study that uses weightlifting as the performance measure than one that uses darts as the performance measure, where individual differences in skill would be more important.

A fourth factor that has not been investigated extensively relates to the variables involved in learning sport psychology techniques. Techniques such as imagery, self-talk and goal setting all require cognitive processing and as such some people will learn them faster than others [56]. Further, some people are intuitive self-taught users of, for example, mood regulation strategies such as abdominal breathing or listening to music who, if recruited to participate in a study investigating the effects of learning such techniques on performance, would respond differently to novice users. Hence, a major challenge when testing the effects of a psychological intervention is to establish suitable controls. A traditional non-treatment group offers one option, but such an approach does not consider the influence of belief effects (i.e., placebo/nocebo), which can either add or detract from the effectiveness of performance interventions [57]. If an individual believes that, an intervention will be effective, this provides a motivating effect for engagement and so performance may improve via increased effort rather than the effect of the intervention per se.

When there are positive beliefs that an intervention will work, it becomes important to distinguish belief effects from the proposed mechanism through which the intervention should be successful. Research has shown that field studies often report larger effects than laboratory studies, a finding attributed to higher motivation among participants in field studies [58]. If participants are motivated to improve, being part of an active training condition should be associated with improved performance regardless of any intervention. In a large online study of over 44,000 participants, active training in sport psychology interventions was associated with improved performance, but only marginally more than for an active control condition [59]. The study involved 4-time Olympic champion Michael Johnson narrating both the intervention and active control using motivational encouragement in both conditions. Researchers should establish not only the expected size of an effect but also to specify and assess why the intervention worked. Where researchers report performance improvement, it is fundamental to explain the proposed mechanism by which performance was enhanced and to test the extent to which the improvement can be explained by the proposed mechanism(s).

## Limitations

Systematic reviews are inherently limited by the quality of the primary studies included. Our review was also limited by the quality of the meta-analyses that had summarized the primary studies. We identified the following specific limitations; (1) only 12 meta-analyses summarized primary studies that were exclusively intervention-based, (2) the lack of detail regarding control groups in the intervention meta-analyses, (3) cross-sectional and correlation-based meta-analyses by definition do not test causation, and therefore provide limited direct evidence of the efficacy of interventions, (4) the extensive array of performance measures even within a single meta-analysis, (5) the absence of mechanistic explanations for the observed effects, and (6) an absence of detail across intervention-based meta-analyses regarding number of sessions, participants' motivation to participate, level of expertise, and how the intervention was delivered. To ameliorate these concerns, we included a quality rating for all included meta-analyses. Having created higher and lower quality groups using a median split of quality ratings, we showed that effects were larger, although not significantly so, in the higher quality group of meta-analyses, all of which were published since 2015.

## Conclusions

Journals are full of studies that investigate relationships between psychological variables and sport performance. Since 1983, researchers have utilized meta-analytic methods to summarize these single studies, and the pace is accelerating, with six relevant meta-analyses published since 2020. Unquestionably, sport psychology and performance research is fraught with limitations related to unsophisticated experimental designs. In our aggregation of the effect size values, most were small-to-moderate in meaningfulness with a handful of large values. Whether these moderate and large values could be replicated using more sophisticated research designs is unknown. We encourage use of improved research designs, at the minimum the use of control conditions. Likewise, we encourage researchers to adhere to meta-analytic guidelines such as PRISMA and for journals to insist on such adherence as a prerequisite for the acceptance of reviews. Although such guidelines can appear as a 'painting by numbers' approach, while reviewing the meta-analyses, we encountered difficulty in assessing and finding pertinent information for our study characteristics and quality ratings. In conclusion, much research exists in the form of quantitative reviews of studies published since 1934, almost 100 years after the very first publication about sport psychology and performance [2]. Sport psychology is now truly global in terms of academic pursuits and professional practice and the need for

best practice information plus a strong evidence base for the efficacy of interventions is paramount. We should strive as a profession to research and provide best practices to athletes and the general community of those seeking performance improvements.

## Supporting information

**S1 Checklist.**
(DOC)

## Acknowledgments

We acknowledge the work of all academics since Koch in 1830 [2] for their efforts to research and promote the practice of applied sport psychology.

## Author Contributions

**Conceptualization:** Marc Lochbaum.

**Data curation:** Marc Lochbaum, Elisabeth Stoner, Tristen Hefner, Andrew M. Lane, Peter C. Terry.

**Formal analysis:** Marc Lochbaum, Peter C. Terry.

**Methodology:** Marc Lochbaum, Elisabeth Stoner, Tristen Hefner, Andrew M. Lane, Peter C. Terry.

**Project administration:** Marc Lochbaum.

**Supervision:** Marc Lochbaum.

**Writing – original draft:** Marc Lochbaum, Elisabeth Stoner, Sydney Cooper, Andrew M. Lane, Peter C. Terry.

**Writing – review & editing:** Marc Lochbaum, Sydney Cooper, Andrew M. Lane, Peter C. Terry.

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
