## [Decision Letter · Decision Letter 0]

7 Dec 2021

PONE-D-21-31186Sport psychology and performance meta-analyses: A systematic review of the literaturePLOS ONE

Dear Dr. Lochbaum,

Thank you for submitting your manuscript to PLOS ONE. After careful consideration, we feel that it has merit but does not fully meet PLOS ONE’s publication criteria as it currently stands. Therefore, we invite you to submit a revised version of the manuscript that addresses the points raised during the review process.

We look forward to receiving your revised manuscript.

Kind regards,

Claudio Imperatori, Ph.D

Academic Editor

PLOS ONE

Journal Requirements:

2. We noted in your submission details that a portion of your manuscript may have been presented or published elsewhere. [I believe our answer is no. In 2017, I spoke about meta-analyses and sport performance at the Kinesiology Conference in Croatia. This conference proceeding is appropriately referenced.] Please clarify whether this conference proceeding was peer-reviewed and formally published. If this work was previously peer-reviewed and published, in the cover letter please provide the reason that this work does not constitute dual publication and should be included in the current manuscript.

Reviewers' comments:

Reviewer's Responses to Questions

**Comments to the Author**

1. Is the manuscript technically sound, and do the data support the conclusions?

Reviewer #1: Yes

Reviewer #2: Yes

2. Has the statistical analysis been performed appropriately and rigorously? 

Reviewer #1: Yes

Reviewer #2: Yes

3. Have the authors made all data underlying the findings in their manuscript fully available?

Reviewer #1: No

Reviewer #2: Yes

4. Is the manuscript presented in an intelligible fashion and written in standard English?

Reviewer #1: No

Reviewer #2: Yes

5. Review Comments to the Author

Reviewer #1: The paper entitled: “Sport psychology and performance meta-analyses: A systematic review of the literature” aimed to synthesize the extant literature to gain insights into the overall impact of sport psychology on athletic performance. The paper is well written and has a great and strong methodology. However, the introduction and discussion are not persuasive enough that the findings make a significant contribution to the literature and could therefore override these limitations. I include some comments below related to this summary for consideration.

1. In relation to the contribution of the study to the literature, I did not get a sense from the article that the findings revealed anything other than what we already know. Please clarified that;

2. The introduction of the paper was very descriptive, it did not situate the current study in literature or highlight what the gap in the literature is that this study is trying to address. At least, the authors should situate better the main purposes of this study;

3. The discussion is very descriptive and any statements about the contribution and conclusions of the study are not new. At least this moment. Please clarified better and justified your choices.

4. Overall, the paper has conditions for be accepted in PLOS ONE, however the authors should clarified the points above.

Reviewer #2: The submitted work presents a very interesting approach to summarize the results of systematic reviews/meta-analysis regarding sport psychology and performance. I must say that it is rare as a reviewer to find a so relevant and well developed study (particularly a review of literature) in which I can add and help so little. The authors are to be commended for the excellent work developed.

Given this, I can make 1 or 2 remarks in some sections, although I do not believe they are needed to ensure a final quality of the developed work. I believe this work can be published as it is, and my comments should only be considered if the authors feel they are noteworthy.

Lines 99 to 102. Given that several examples were presented before (e.g., journals), why the inclusion of only one book? Several examples could be given here, thus maintaining the line of reasoning presented before.

In method, why report PRISMA 2009, 2015 and 2020 guidelines? As stated in the Page et al (2020) reference used: "The PRISMA 2020 statement replaces the 2009 statement and includes new reporting guidance that reflects advances in methods to identify, select, appraise, and synthesis studies". Won't the 2020 reference be enough?

As a last remark, I wonder if a discussion (or a comment in the discussion/limitations) regarding mood, and particularly POMS, is needed. In this work and in some of the cited works (e.g., Lochbaum et al., 2021, EJIHPE) no discussion regarding the issues of POMS as an assessing tool for mood is presented. As mentioned by several researchers (e.g., Ekkekakis, 2013), POMS do not assess mood, at least not in a global domain. This do not impact directly this work, as generally only each of the six distinct states are explored. However, when interpreting figure 2 and extracting mood results, perhaps some clarification would frame the readers on this issues and respective interpretation of results.

Ekkekakis, P. (2013). The measurement of affect, mood and emotion. Cambridge University Press.

I am sorry I can not help any further with my comments. Thank you for your work.

Best regards

6. PLOS authors have the option to publish the peer review history of their article (what does this mean?). If published, this will include your full peer review and any attached files.

Reviewer #1: No

Reviewer #2: **Yes: **Diogo S. Teixeira

---

## [Author Response · Author response to Decision Letter 0]

13 Dec 2021

Response to Reviewers

 Thank you to both reviewers for taking time to review and comment on our manuscript. We addressed all comments.

Reviewer's Responses to Questions

1. Is the manuscript technically sound, and do the data support the conclusions?

Reviewer #1: Yes

Reviewer #2: Yes

Author response: Thank you to the reviewers for their positive comments.

2. Has the statistical analysis been performed appropriately and rigorously? 

Reviewer #1: Yes

Reviewer #2: Yes

Author response: Thank you to the reviewers for their positive comments.

3. Have the authors made all data underlying the findings in their manuscript fully available?

Reviewer #1: No

Reviewer #2: Yes

Author response: All pertinent data are found in Table 1 – 2 and in Figure 1.

4. Is the manuscript presented in an intelligible fashion and written in standard English?

Reviewer #1: No

Reviewer #2: Yes

Author response: Reviewer 1’s concerns have been addressed below.

5. Review Comments to the Author

Reviewer #1

The paper entitled: “Sport psychology and performance meta-analyses: A systematic review of the literature” aimed to synthesize the extant literature to gain insights into the overall impact of sport psychology on athletic performance. The paper is well written and has a great and strong methodology. However, the introduction and discussion are not persuasive enough that the findings make a significant contribution to the literature and could therefore override these limitations. I include some comments below related to this summary for consideration.

• Author response: We have amended the paper to address the three concerns below.

Comment 1. In relation to the contribution of the study to the literature, I did not get a sense from the article that the findings revealed anything other than what we already know. Please clarified that;

• Author response: We have expanded on the gap in the knowledge that we addressed on lines 115-121 on the revised manuscript. 

Comment 2. The introduction of the paper was very descriptive, it did not situate the current study in literature or highlight what the gap in the literature is that this study is trying to address. At least, the authors should situate better the main purposes of this study;

• Author response: Currently, sport psychology practitioners wishing to use evidence-based strategies are faced with inconsistent evidence about the efficacy of sport psychology techniques. Our paper addresses this inconsistency by assessing the effectiveness of techniques collectively. This is explained on lines 115-121 and with some small modifications on lines 125-128.

Comment 3. The discussion is very descriptive and any statements about the contribution and conclusions of the study are not new. At least this moment. Please clarified better and justified your choices.

• Author response: As suggested, a stronger summary of the contribution of the paper is provided on lines 371-375. We would also argue that the recommendations section for improvements to future studies also represents a significant contribution to the body of knowledge. If the information provided is already well known, as the reviewer suggests, then we would question why previous investigators have not implemented it in their studies.

Comment 4. Overall, the paper has conditions for be accepted in PLOS ONE, however the authors should clarified the points above.

• Author response: We thank you for your comments, which have served to improve our paper.

Reviewer #2

The submitted work presents a very interesting approach to summarize the results of systematic reviews/meta-analysis regarding sport psychology and performance. I must say that it is rare as a reviewer to find a so relevant and well developed study (particularly a review of literature) in which I can add and help so little. The authors are to be commended for the excellent work developed.

• Author response: Many thanks for your extremely positive comments.

Comment 1. Given this, I can make 1 or 2 remarks in some sections, although I do not believe they are needed to ensure a final quality of the developed work. I believe this work can be published as it is, and my comments should only be considered if the authors feel they are noteworthy.

Lines 99 to 102. Given that several examples were presented before (e.g., journals), why the inclusion of only one book? Several examples could be given here, thus maintaining the line of reasoning presented before.

• Author response: As suggested, we have added some additional references to books on lines 99-104 and added them to the reference list on lines 523-524 and 527-529.

Comment 2. In method, why report PRISMA 2009, 2015 and 2020 guidelines? As stated in the Page et al (2020) reference used: "The PRISMA 2020 statement replaces the 2009 statement and includes new reporting guidance that reflects advances in methods to identify, select, appraise, and synthesis studies". Won't the 2020 reference be enough?

• Author response: As suggested, we have removed reference to the PRISMA guidelines published in 2009 and 2015.

Comment 3. As a last remark, I wonder if a discussion (or a comment in the discussion/limitations) regarding mood, and particularly POMS, is needed. In this work and in some of the cited works (e.g., Lochbaum et al., 2021, EJIHPE) no discussion regarding the issues of POMS as an assessing tool for mood is presented. As mentioned by several researchers (e.g., Ekkekakis, 2013), POMS do not assess mood, at least not in a global domain. This do not impact directly this work, as generally only each of the six distinct states are explored. However, when interpreting figure 2 and extracting mood results, perhaps some clarification would frame the readers on this issues and respective interpretation of results.

Ekkekakis, P. (2013). The measurement of affect, mood and emotion. Cambridge University Press.

• Author response: It was not our intent to critique the construct validity of the measures used in the meta-analyses we reviewed. Nevertheless, as suggested, we have added a note that the POMS and its derivatives do not measure all aspects of the global domain of mood (see lines 364-366).

I am sorry I cannot help any further with my comments. Thank you for your work.

Best regards

• Author response: We are delighted to know that you thought so highly of our paper.

---

## [Decision Letter · Decision Letter 1]

19 Jan 2022

Sport psychology and performance meta-analyses: A systematic review of the literature

PONE-D-21-31186R1

Dear Dr. Lochbaum,

We’re pleased to inform you that your manuscript has been judged scientifically suitable for publication and will be formally accepted for publication once it meets all outstanding technical requirements.

Kind regards,

Claudio Imperatori, Ph.D

Academic Editor

PLOS ONE

Additional Editor Comments (optional):

Reviewers' comments:

Reviewer's Responses to Questions

**Comments to the Author**

1. If the authors have adequately addressed your comments raised in a previous round of review and you feel that this manuscript is now acceptable for publication, you may indicate that here to bypass the “Comments to the Author” section, enter your conflict of interest statement in the “Confidential to Editor” section, and submit your "Accept" recommendation.

Reviewer #1: All comments have been addressed

Reviewer #2: All comments have been addressed

2. Is the manuscript technically sound, and do the data support the conclusions?

Reviewer #1: Yes

Reviewer #2: Yes

3. Has the statistical analysis been performed appropriately and rigorously? 

Reviewer #1: Yes

Reviewer #2: Yes

4. Have the authors made all data underlying the findings in their manuscript fully available?

Reviewer #1: No

Reviewer #2: Yes

5. Is the manuscript presented in an intelligible fashion and written in standard English?

Reviewer #1: Yes

Reviewer #2: Yes

6. Review Comments to the Author

Reviewer #1: No more comments. The authors adressed all of my previous comments. Therefore, the manuscript must be accepted for publication. Congratulations.

Reviewer #2: (No Response)

7. PLOS authors have the option to publish the peer review history of their article (what does this mean?). If published, this will include your full peer review and any attached files.

Reviewer #1: No

Reviewer #2: **Yes: **Diogo S. Teixeira

---

## [Editor Report · Acceptance letter]

25 Jan 2022

PONE-D-21-31186R1 

Sport psychology and performance meta-analyses: A systematic review of the literature 

Dear Dr. Lochbaum:

I'm pleased to inform you that your manuscript has been deemed suitable for publication in PLOS ONE. Congratulations! Your manuscript is now with our production department. 

Kind regards, 

on behalf of

Dr. Claudio Imperatori 

Academic Editor

PLOS ONE